# Built-Up Growth Impacts on Digital Elevation Model and Flood Risk Susceptibility Prediction in Muaeng District, Nakhon Ratchasima (Thailand)

**Patiwat Littidej * and Nutchanat Buasri**

Department of Geo-informatics, Faculty of Informatics, Mahasarakham University,
Mahasarakham 44150, Thailand
* Correspondence: patiwat.l@msu.ac.th

**Abstract:** The transformation of land-use and land cover in Nakhon Ratchasima province, Thailand has rapidly changed over the last few years. The major factors affecting the growth in the province arise from the huge expansion of developing areas, according to the government's development plans that aim to promote the province as a central business-hub in the region. This development expansion has eventually intruded upon and interfered with sub-basin areas, which has led to environmental problems in the region. The scope of this study comprises three objectives, i.e., (i) to optimize the Cellular Automata (CA) model for predicting the expansion of built-up sites by 2022; (ii) to model a linear regression method for deriving the transition of the digital elevation model (DEM); and (iii) to apply Geographic Weighted Regression (GWR) for analyzing the risk of the stativity of flood areas in the province. The results of this study show that the optimized CA demonstrates accurate prediction of the expansion of built-up areas in 2022 using Land use (LU) data of 2-year intervals. In addition, the predicting model is generalized and converged at the iteration no. 4. The prediction outcomes, including spatial locations and ground-water touch points of the construction, are used to estimate and model the DEM to extract independent hydrology variables that are used in the determination of Flood Risk Susceptibility (FRS). In GWR in the research called *FRS-GWR*, this integration of quantitative GIS and the spatial model is anticipated to produce promising results in predicting the growth and expansion of built-up areas and land-use change that lead to an effective analysis of the impacts on spatial change in water sub-basin areas. This research may be beneficial in the process of urban planning with respect to the study of environmental impacts. In addition, it can indicate and impose important directions for development plans in cities to avoid and minimize flood area problems.

**Keywords:** flood risk susceptibility; *FRS-GWR* modeling; built-up growth prediction; Thailand

## 1. Introduction

Land-use in areas typically involves dynamical processes that develop land which changes over time, based on the evolution of the economy, society, and population in the areas [1]. Population growth results and increases the demands and expansion of land-use in such areas. As a consequence, this fabricates land-use planning (especially the expansion of residential and building areas), which leads to substantial replacements owing to restricted resources [2–4]. For example, there is an inevitable rational in some areas to replace dwelling and development zones in agricultural lands [5]. In the last five years, the increased development and expansion of built-up areas has impacted dramatically on land-use and land-cover characteristics. In Nakhon Ratchasima (urban zone), Thailand, large areas of agricultural land have essentially been transformed for development, housing, and dwelling

zones according to the current government's master plans. The plans aim to increase the potential transportation capabilities in order to cope with the expansion of the cities in the country by launching a number of construction projects in the province, i.e., a Highway (no. 6), a double-track train line, and a high-speed train project [6]. As a result, this eruptive non-planned land-use change can cause significant environmental problems, for instance, water logging, traffic, and buildup of solid waste [7–9].

Land-use change is one of the factors that can impact on flow accumulation in water networks, including flow accumulation, flow length, and areas of sub-basins. These flow accumulations are sensitive hydrological factors that play a vital role in determining the change of the DEM in the areas [10,11]. The impact of urban growth is one of the significant land-use changes affecting surface runoff within the catchment area [12]. Urban expansion also leads to the removal of trees and vegetation, causing a decrease in evapotranspiration. The construction of roads and culverts has effects which may include reduction of infiltration, decline in the groundwater table, increased surface runoff, and a reduction of base flows [13–15]. Built-up areas lead to an increase in DEM surfaces which in turn reduces the runoff concentration time. Accordingly, higher peak discharge rates occur sooner after rainfall in the catchment area [16]. In addition, the runoff volume and potential flood risk greatly increases [17–19]. Converting agricultural land to built-up arcades, for example, may require land-fill work to level the height of the land up to at least 2–3 m before construction can be carried out to minimize water logging risks. This land-fill can ultimately interfere and block the flow accumulation, thus incurring a change in the flow directions. As a result, rain-water cannot be drained to main local rivers effectively when a large volume of rain is received; this subsequently creates water logging in the area [20–22]. Land height in sub-basin regions is an important factor that imposes a substantial flow length and flow accumulation. If the density of flow length and the number of accumulated flows are small compared with the size of the sub-basins, the performance of the water drain process will become degraded and result in water-logging [23,24]. Prediction of the growth and expansion of these development areas can be utilized to generate an accurate transition DEM map [25,26]. In addition, it can be used to analyze quantitative hydrological factors. However, land-use modeling is more complicated as it needs to integrate different pieces of contextual information, such as spatial information and some independent variables that drive land-use change [27–31]. The factors that make DEM different in each area are also dependent on the nature of the style of the land-use, and the social and economic form. In recent studies, there has been no research which has been used to increase the height of the area with such local factors.

Predicting land-use and land-cover change using mathematic models and geo-information systems is one of the popular methods as it can visualize outcomes in spatial information patterns, which are not limited to the quantitative information available in other conventional methods. Land-use information is fundamental to analyzing the changes of land coverage by analyzing the various spatial software [31]. This work applies a wide range of spatial models, such as Geomod2 [32], CA, SLEUTH Urban Growth [33], Lucas [34], and LTM [35]. Each model has its own characteristics and processes. However, they function in the same capacities, which are used to model the change of land- use and land-cover. Additionally, each technique is applicable to different geo-information software. CLUE-S can be used together with ArcGIS, QGIS, and IDRISI. Lucas, on the other hand, can be coupled with GRASS and Geomod2, which can be applied with IDRISI. Likewise, CA is a sub- module available in IDRISI. Therefore, there are a number of tools that can be applied to model the change of land-use and land-cover. Integration between land-use forecast models and the DEM modeling can be applied to the modeling to analyze the risk of flood disaster.

This research applies CA integrated with optimization schemes to predict built-up growth in areas that have experienced rapid growth (construction and development). The prediction model is used, thereafter, to model the change of DEM in order to precisely locate and estimate the change of DEM according to the growth of buildings in the areas. In addition, it is utilized in modeling GWR for analyzing sensitive areas and areas that are at risk to water flooding resulting from the growth of construction in these areas. This integration of quantitative and spatial models is anticipated to

produce promising results in predicting spatial change in water sub-basin areas. This study is aimed at (i) optimizing the CA model for predicting the expansion of built-up polygon in 2022; (ii) modeling a linear regression method for deriving the transition of DEM and hydrological characteristic analysis for independent variable extraction; and (iii) applying the GWR model for analyzing the risk of the stativity of flood areas.

## 2. Research Methodology

### 2.1. Study Area

Amphoe Mueang Nakhon Ratchasima, Thailand has an area of approximately 645.84 sq. km There are 22 communities in the city, "Korat", which is located in the center of the province. In 2014, land-use and land-cover in the city was deliberately divided into five main types (as illustrated in Figure 1 and Appendix A Table A1), i.e., (i) agricultural areas (385.01 sq. km 59.61%); (ii) forest (7.12 sq. km, 1.10%); (iii) water resources (7.45 sq. km, 1.15%); (iv) built-up and dwellings (202.68 sq. km, 31.40%); and (v) others (43.58 sq. km, 3.33%) respectively. Geographically, this study area is located in the Lamtaklong basin (WGS 1984 UTM Zone 48 N (X = 169,957 to 202,237) E and (Y = 1,637,208 to 1,675,772) N). Spatial level is between 164 to 277 m. Lamtaklong River is the Main River and cuts through the center of the province. This river is the major source that supports the water- draining flowing of the floodway from west to east across the province. From a hydrological perspective, the majority of areas in the northeast are lowland areas, which sit lower than those in the east. As the city is situated close to Lamtaklong River and there is an ongoing construction project in the area, including the endpoint of the highway and bypass road (Nakhon Ratchasima–Khon Kean) congestion, the area is susceptible to a water-logging problem. In some situations, water cannot drain as run-off efficiently after rainfall for 20 min. This non-planned expansion of built-up and developing areas causes land-filling (of up to 2 m from regular levels), and a change of land-use conditions, resulting in tremendous problems with the sub-basins and from flow accumulation.

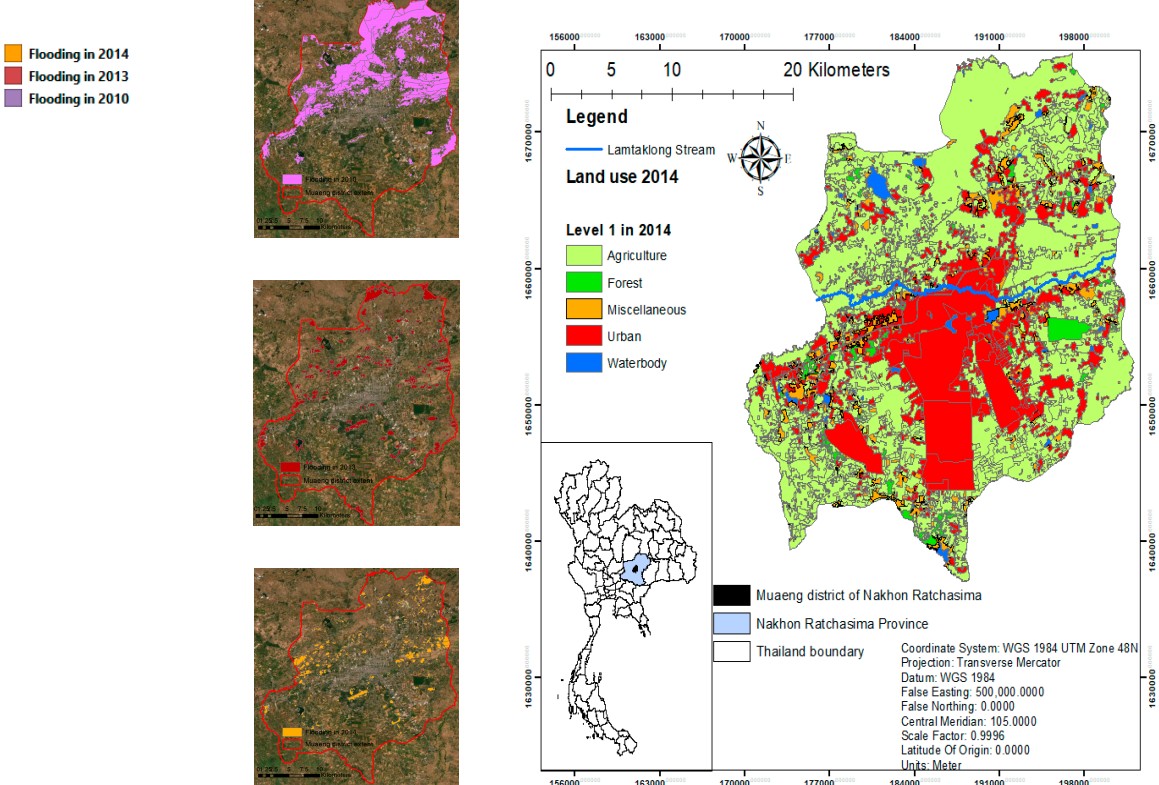

**Figure 1.** Significant floods in 2010, 2013, and 2014 covered in land-use/land cover (LU/LC) of Mueang district extent, Nakhon Ratchasima province derived using visual interpretation in December 2014.

To alleviate these problems, therefore, this research devised three objectives (depicted in Figure 2) as follows. Firstly, the prediction of built-up areas by 2022 using CA is performed using high-resolution IKONOS data (downloaded from Google Earth Pro). The image data was collected in December 2014, 2016, and 2018, as these times are at the end of the annual land-use change with less cloud scattering hindering the visualization of the data. The model will project predicted data to produce a prediction map of built-up and development areas by 2022. However, this predictive model cannot be appropriately applied to this study area due to the fact that the change of land-use and the expansion of built-up and construction in the area is exponentially eruptive-resulting from a number of the government's mega projects. Therefore, optimization techniques are implemented over CA.

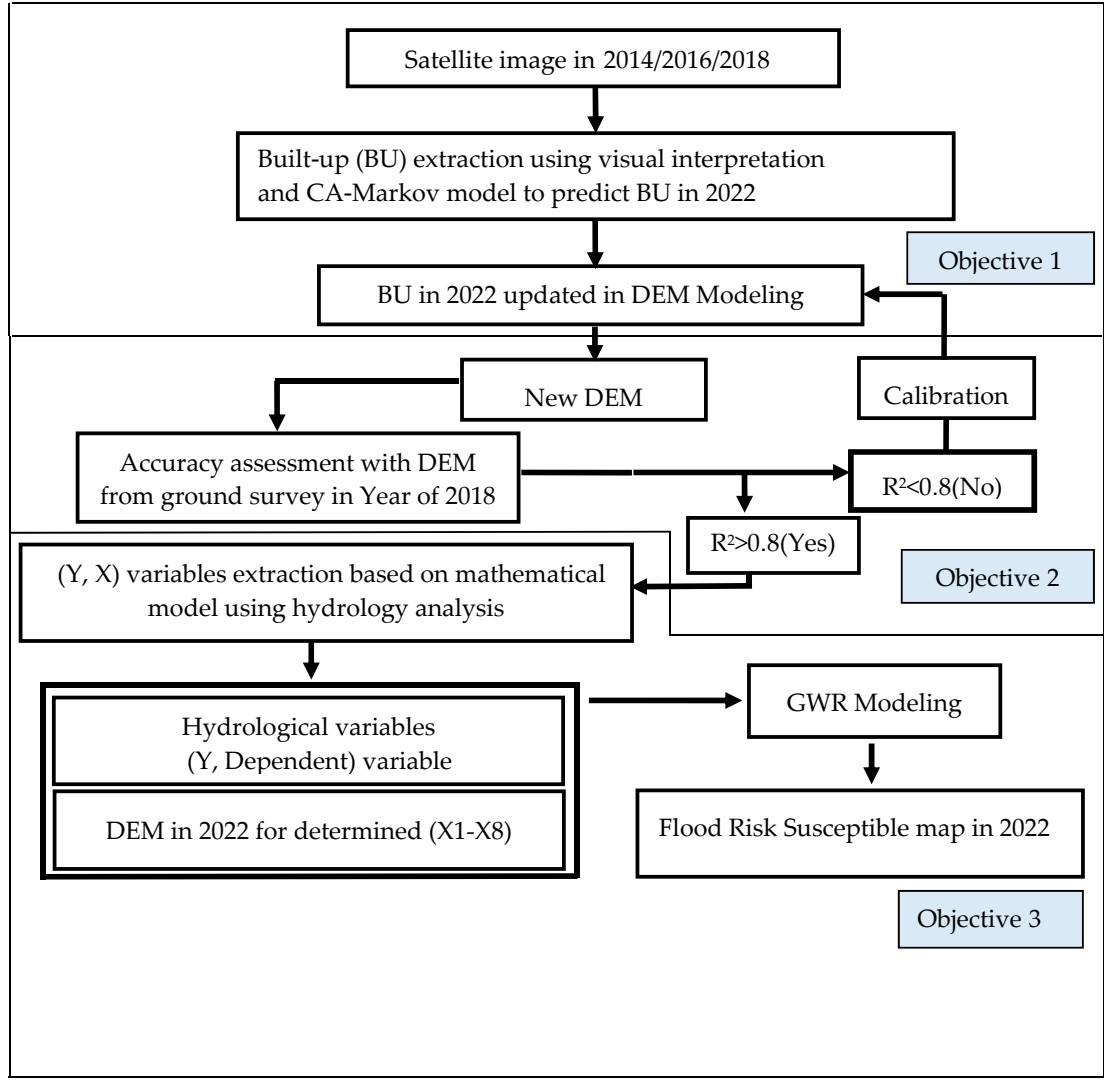

**Figure 2.** Framework of the study.

*2.2. Research Approach*

The procedure for the main operation has three phases: 1. Translation of land data from 3 satellite image data over time, then taking the data to forecast land-use in 2022. The next step is to bring out the projected year land utilization data from the model to calibrate and use to generate numerical height information. This process uses linear models and independent variables that involve the height in the study area. The final step is the introduction of the projected altitude data based on the change from the building's rise as a basis to create a combination of variables based on the independent variables in the GWR model. This thorough research process can be described in spatial data as follows:

An iterative procedure was applied to generate a number of built-up features in the study area. Their polygon was used in the new DEM modeling. Secondly, a new DEM manipulation was generated from the built-up area in 2022 via DEM linear modeling. The DEM prediction model was adjusted to have an accuracy greater than 0.8, and after that it is used as DEM in 2022. DEM in the base year and Euclidean distance of built-up to streamline are used in the model. Third, the updated heights are used to analyze the hydrologic characteristics to create a flood areas index (FAI) and Flood Risk Susceptible (FRS) model. The DEM in 2022 can generate independent variable data in eight different types of sub-basin. The GWR model tests the ability to predict FRS from the indexes $R^2$ and AIC. The factors are screened only using the independent variables that influence the sensitive area and the risk of flooding from the increased number of buildings.

### 2.3. Built-Up Growth Prediction Model

Many studies and researches on prediction of urban growth using different models have been carried out throughout the world. In this study, we select two models for urban growth prediction: The Cellular Automata (CA) model using CA-Markov in IDRISI software package and the Geographic weighted regression (GWR) model which are summarized herein. CA and GWR models are selected to predict built-up growth, the results of which are then compared with the interpreted built-up polygon in 2018.

In references by Benenson and Suwit [36,37] it was said, after that, the model which provides greater accuracy will be used for built-up growth prediction in 2022. The main tasks concern urban growth prediction in 2022 using CA-Markov, selection of the optimum iteration model for built-up growth prediction in the future and prediction of urban growth in 2018 and 2022 as well as similar approaches conducted by Benenson and Suwit [36,37]. Cellular automata are dynamic models discrete in time, space and state. A simple cellular automata A is defined by a lattice (*L*), a state space (*Q*), a neighborhood template $\delta$ and a local transition function (*f*):

$$A = (L, Q, \delta, f) \tag{1}$$

Each cell of *L* can be in a discrete state out of *Q* according to Benenson and Suwit [36,37]. The cells can be linked in different ways. Cells can change their states in discrete time-steps. Usually, cellular automata are synchronous, i.e., all cells change their states simultaneously. The fate of a cell is dependent on its neighborhood and the corresponding transition function *f* [38,39]. The formal definition of Markov processes is very close to that of CA. The Markov process is considered in discrete time and characterized by variables that can be in one of *N* states from $S = \{S1, S2, \dots, S_N\}$. The set *T* of transition rules is substituted using a matrix of transition probabilities (*P*) and this is reflective of the stochastic nature of the process:

$$P_{ij} = \begin{bmatrix} P_{11} & P_{12} & P_{1n} \\ P_{21} & P_{22} & P_{2n} \\ \dots\dots & \dots\dots & \dots\dots \\ P_{n1} & P_{n2} & P_{nn} \end{bmatrix} \tag{2}$$

where $P_{ij}$ is the conditional probability that the state of a cell at moment $t + 1$ will be $S_j$, given it is $S_i$ at moment *t*:

$$prob(S_i \rightarrow S_j) = P_{ij} \tag{3}$$

Ref. by Suwit [37] said that the Markov process as a whole is given by a set of status *S* and a transition matrix *P*. By definition, in order to always be 'in one of the states' for each *i*, the condition $\sum_j P_{ij} = 1$ should hold [39]. The research made the appropriate configuration of the iteration of CA model in IDRISI for built-up prediction by 2022.

### 2.4. Geographically Weighted Regression (GWR) Model of Flood Risk Susceptible (FRS) Prediction

Geographically Weighted Regression [40,41] is used to combine data in each point of observation into a regression model using a series of distance-related weights. The relationship between growth built-up area and location characteristics for a particular point, for example, is given a higher weight than for points further away from that point. The GWR regression model for Flood area predicted ($F_j$) to set as dependent variable, (*Y*) prediction in Nakhon Ratchasima municipality is shown in the following Equation (4).

$$\begin{aligned} F_j(u_i v_i) = &\ \beta_0(u_i v_i) + \beta_1 X_1(u_i v_i) + \beta_2 X_2(u_i v_i) + \beta_3 X_3(u_i v_i) + \\ &\ \beta_4 X_4(u_i v_i) + \beta_5 X_5(u_i v_i) + \beta_6 X_6(u_i v_i) + \beta_7 X_7(u_i v_i) + \beta_8 X_8(u_i v_i) + \varepsilon_i \end{aligned} \tag{4}$$

where $\beta_0$ is the intercept term, $\beta_1, \beta_2, \beta_3, \beta_4, \beta_5$, and $\beta_6$ are spatially varying coefficients of the Normalized Digital Elevation Model Index (*NDEMI*, $X_1$), Normalized Contour Index (*CLI*, $X_2$), Area of Built-up Index (*ABUI*, $X_3$), Density of Built-up ($D_j$, $X_4$), Curvature Index (*CI*, $X_5$), Slope length Index (*SLI*, $X_6$), perimeters ($P_j$, $X_7$), and sub-basin area ($S_j$, $X_8$) attributes respectively, and $\varepsilon_i$ is an error term at point $i$, ($u_i v_i$) representing the coordinates of the $i^{\text{th}}$ point in study extent [41]. The GWR model is modeled using the GWR 3.0 software package which allows the use of a variety of calibration techniques to specify regression weights and to optimize bandwidth parameters. In this study, a fixed defined kernel with a bi-square function (in which the bandwidth was determined through the minimization of the Akaike Information Criterion (AIC) [40,41] was used. The reason for this is that the points in the spatial unit of analysis used are in regular and equal sizes. Monte Carlo tests [40,41] were also carried out to set the significance of the spatial variability in the local parameter estimates. Independent variable weight values are indicators of which independent variables will influence flood risk. The coefficients of independent variables have both positive and negative relationships affecting the change of the flood areas. A series of pre-tested, independent variables provides a satisfactory result in terms of statistical index value, which is defined as an independent variable to use the Flood Risk Susceptibility (FRS) prediction in each sub-basin.

The map showing the flood risk area size may not reflect the severity of the spatial unit of each sub-basin. Therefore, it is necessary to create an additional model to describe the sensitivity of the flood. Flood Area Index (FAI) was used as a major input in the Flood Risk Susceptibility (FRS) model and shown as a probability in the term between 0–100 in the following Equation (5) as FAI and Equation (6) as FRS, respectively.

Many of the previously flooded areas act as indicators of the frequency and severity of flooding in those areas. Flood information in the past was interpreted from the satellite imagery by Geo-Informatics and Space Technology Development Agency (Public Organization), (GISTDA). Flood data from 2010–2016 was able to be used to establish a Flood Area Index (FAI Dependent variable, (Y)) as shown in Equation (5)

$$F_j = \frac{\sum\limits_{i=1}^{n} F_{ijk}}{N_k} \tag{5}$$

where $F_j$ is Flood Area Index of each sub-basin any $j$, $\sum\limits_{i=1}^{n} F_{ijk}$ is the summation of flooding in the cell any $j$ of each sub-basin any $j$ in the year any $k$ and $N_k$ is a number of flooding the cell in the year any $k$.

$$FRS_j = \frac{F_j}{S_j} \times 100 \tag{6}$$

where $FRS_j$ is flood risk susceptibility in 2022 of each sub-basin any $j$ where the closer the value is to 1, the greater the risk (ranged as 0–1) and $F_j$ is Flood Area Index (FAI) predicted in each sub-basin of any $j$ (sq. km) and $S_j$ is the area of each sub-basin of any $j$ (sq. km).

## 2.5. Independent Variables Modeling from Hydrological Characteristics

This study begins by undertaking a Digital Elevation Model (DEM) modeling of a numerical height to predict the height of the space in the desired future year. These height data are extracted as an index of the independent variables to be used to create the GWR model. This also uses the flood data in the past that occurred in the leading area index of Flood Area Index (FAI). The FAI index is a dependent variable in the GWR model and is extracted to a boundary of each sub-basin and displays the details of the various indices.

### 2.5.1. DEM Prediction Modeling

A new DEM in each sub-basin of each following year is derived from the linear model following Equation (6). The height of DEM on the base-year and the distance of built-up polygons from the stream are input into the model. The altitude change in the study area is a major factor affecting the independent variables mentioned above. The coefficient of the independent variable is used to identify the relationship between the distances from the stream to the predicted altitude as shown in Equation (7)

$$D_{i(n+1)} = D_{i(n)} \pm L_{is} \tag{7}$$

where $D_{i(n+1)}$ is elevated point at any $i$ in next ($n$ + 1) year (meters), $D_{i(n)}$ is elevation point of cell any $i$ in $n$ year (meters,) and $L_{is}$ is a distance between the center point of built-up polygons of feature any $i$ to the closet Lamtaklong stream line feature $s$ (meters).

### 2.5.2. Normalized Digital Elevation Model Index (NDEMI, X1)

The DEM index is made to analyze the difference in the average altitude in the sub-basin area, with the assumption that if any of the sub-basin areas are different, then there is a greater risk of flooding. However, the index only applies to this analysis of risk-prone areas. Flooding may not be able to identify the total area risk due to the difference in the altitude of the area, resulting in the water flowing into other subgroups and quickly venting. Therefore, the risk analysis of flooding is necessary to be used in conjunction with other independent variables. The output of this index is adjusted to a standard in the range of 0–1. The higher the altitude, the closer the value is to 1, and the lower value will approach 0. The index guideline makes it possible to measure the difference in the same standard of that sub-basin. The index shows the calculation of Equation (8).

$$NDEMI_j = \frac{\sum\limits_{i=1}^{n} \left[ \frac{E_i - E_{\min}}{E_{\max} - E_{\min}} \right]}{\sum\limits_{i=1}^{n} N_{ij}} \tag{8}$$

where the $NDEMI_j$ index is the normalized digital elevation model of the sub-basin any $j$, is the proportion of the standard average of the difference in height. $E_i$ is elevation point of any cell $i$, $E_{\max}$ is the highest elevation point of each sub-basin any $j$ (meters), $E_{\min}$ is the lowest elevation point of each sub-basin any $j$ (meters) and $\sum\limits_{i=1}^{n} N_{ij}$ is the summation of the number of elevation points of any cell $i$ in each sub-basin any $j$.

### 2.5.3. Normalized Contour Index (CLI, X2)

The Altitude line length index is calculated based on the multiples of the line length at different heights per size of each area in any sub-basin. A high index value represents the difference in the altitude of a basin. A lower index value indicates that the basin is smooth, as shown in Equation (9).

$$CLI_j = \frac{\sum\limits_{i=1}^{n} C_k L_i}{P_j} \tag{9}$$

where $CLI_j$ is the normalized contour index of sub-basin any $j$. It is the proportion of the sum product of a height level and the length of contour lines per any perimeter of sub-basin any $j$, $C_k$ is a level of contour line any $k$ in sub-basin any $j$, $L_i$ is a length of contour line any $i$ of contour line any $k$ in sub-basin any $j$ and $P_j$ is the perimeter of sub-basin any $j$.

2.5.4. Area of Built-Up Index (ABUI, X3)

This is the proportion of the summation of the built-up area within sub-basin any $j$ per the extent of sub-basin any $j$ (sq. km). The index is used to analyze the larger and smaller built-up space. The increase in the building area makes the flow accumulation and the water flow direction in the basin area more likely to change than the area that rarely changes, as shown in Equation (10).

$$ABUI_j = \frac{\sum\limits_{i=1}^{n} ABUI_i}{S_j} \tag{10}$$

where $ABUI_j$ is the area of built-up Index of sub-basin any $j$, $\sum\limits_{i=1}^{n} ABUI_i$ is the summation of the built-up area in the polygon any $i$ of each sub-basin any $j$ and $S_j$ is the area of sub-basin any $j$ (sq. km).

2.5.5. Density of Built-Up ($D_j$, X4)

This index is similar to the indexes that are referenced above. However, the analysis of the water path change from the built-up area alone may not be sufficient to predict the flood risk susceptible area. In addition, the number of buildings is rapidly increasing and the boundaries are becoming close to one another, causing the water flow to be discontinued and the water is subsequently becoming trapped in a sub-watershed area that looks like this. Using this independent variable with an index ($ABUI$), it is possible to refer to the level of growth of buildings associated with flooding in the sub-basin, as shown in Equation (11).

$$D_j = \frac{\sum\limits_{i=1}^{n} N_i}{S_j} \tag{11}$$

where $D_j$ is built-up density of sub-basin any $j$, $N_i$ is a number of built-up areas at the center point of polygon any $i$ within sub-basin any $j$ and $S_j$ is the area of sub-basin any $j$ (sq. km).

2.5.6. Curvature Index (CI, X5)

In the calculation of areas of the surface curvature, the convex is curved, textured, inverted, or has an embossed appearance, and the concave has a curved, textured, rounded appearance or a puddle. Convex-style areas pose a risk of less flooding than a concave surface, if the area appears flat, there is a value of two areas approaching the center. The index value of any basin is even more vulnerable and susceptible to flooding less than the low index area, as shown in Equation (12).

$$CI_j = \frac{\sum\limits_{i=1}^{n} A_{convex(i)}}{S_j} \tag{12}$$

where $CI_j$ Curvature Index of sub-basin any $j$, $A_{convex(i)}$ is the area of Convex-style areas of polygon any $i$ within sub-basin any $j$ (sq. km) and $S_j$ is the area of sub-basin any $j$ (sq. km).

2.5.7. Slope Length Index (SLI, X6)

The index is calculated from the proportion of the sum of the slope length per each perimeter in the sub-basin. This index is used to analyze high-index values, the high and long slopes in the

sub-basin area, allowing water to accumulate in the lowest possible areas while the risk of detention and drainage is low, as shown in Equation (13).

$$SLI_j = \frac{\sum\limits_{i=1}^{n} SL_i}{P_j} \tag{13}$$

where $SLI_j$ is Slope length Index of sub-basin any $j$, $SL_i$ is a length of slope of polyline any $i$ within sub-basin any $j$ (km) and $P_j$ is the perimeter of sub-basin any $j$.

In addition to the previously-displayed indexes, there is also an independent variable that is used as an important piece of data to determine the index values, such as the perimeters where $(P_j, X_7)$ is a calculation of the length of the circumference of the sub-basin, and the sub-basin area where $(S_j, X_8)$ is a calculation of the area of the sub-basin, where both of these values can be calculated from the mathematical principles of the spatial relationship.

## 3. Results and Discussion

### 3.1. Optimal Iterations of CA-Markov Model for Built-Up Growth Prediction in 2022

The results of interpreting satellite imagery from the years 2014, 2016, and 2018 using the interpretation method interpreted visually (Visual interpretation) based on the composition of the interpretation consisting of the shape (Shape), size (Pattern), the intensity of colors, and colors (Tone and color), the texture (Shadow) location, and the association (Site) and its relevance are classified into five categories, including; (1) community areas and buildings: areas with all types of buildings and residential trade zones as well as government offices and transportation routes; (2) Miscellaneous areas including open space; (3) other agricultural areas such as areas, fields, garden areas, etc.; (4) water source areas that are both natural and man-made water sources; and (5) forest areas, including natural forest areas and planted forests. The results of the 2016 satellite image interpretation found that: (1) the Built-up area comprised a total of 209.45 sq. km (32.43%); (2) the miscellaneous area was 40.25 sq. km (6.25%); (3) the water source area was 7.06 sq. km (1.09%); (4) the forest area was 7.03 sq. km (1.08%); and (5) the agricultural area was 382.05 sq. km (59.15%). The effect of interpreting satellite imagery from 2018 found that: (1) the Built-up area was 225.88 sq. km (34.97%); (2) the miscellaneous area was 36.47 sq. km (5.65%); (3) the water source area was 6.91 sq. km (1.07%); (4) the forest area was 6.52 sq. km (1.01%); and (5) the agricultural area was 370.06 sq. km (57.30%), respectively, and all interpretation result information is shown in the Appendix A Tables A1–A10.

The transition matrix for built-up prediction in 2022 is shown in Table A10 and the dispersion position shown in Figure 3. The result of the 2022-year satellite image translation found that: (1) the Built-up area was 242.85 sq. km (37.6%); (2) the miscellaneous area was 27.49 sq. km (4.26%); (3) the water source area was 6.97 sq. km (1.08%); (4) the forest area was 6.22 sq. km (0.96%); and (5) the agricultural area was 362.31 sq. km (56.1%). Building and road areas tend to increase every year. The main types of soil cover that are transformed into building areas are: Agricultural areas and water sources mostly included in some miscellaneous areas. The water source area is being continually reduced due to the fact that the land is reclaimed in the original water source to make way for constructions, such as housing projects, housing, and other types of homes, as the position of change in this way is formed along the road, avoiding the city and the area near the end of the Highway (main road number-6). Miscellaneous areas are relatively constant, but there is a slight decline shown in the year 2018. This soil coverage data layer is converted into raster data in the form of a file extension (.RST) to import into a raster format. The forecast growth of built-up areas is conducted with CA-Markov models using the IDRISI version 15.0 to create a matrix of changes and analyze the optimal iteration and simulate land utilization changes.

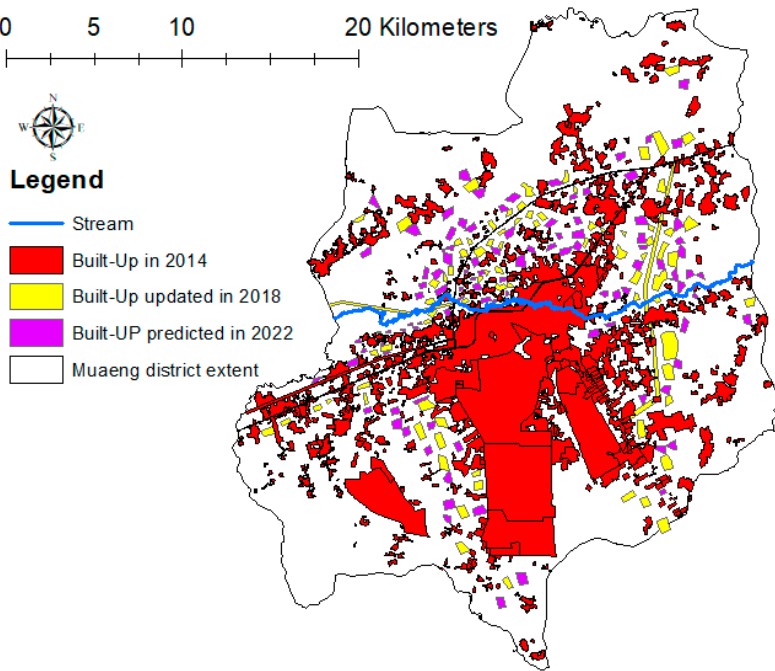

**Figure 3.** Built-up areas in 2014 and 2018 and built-up growth in 2022.

The calculation of a matrix of the CA-Markov model is different in space. This research configured the change matrix (all results of the transition matrix are shown in Tables A1–A10.) and the range of satellite imagery during the year 2014/2016/2018. Selecting a range of images used to create a transition matrix is a 2-year-old pitch, as it requires satellite imagery to show the expansion pattern of a uniform building, and the buildings are scattered with similar shapes and orientations. The model can be used to predict the position of the buildings that will be built in future years according to the assumptions of the duration of the satellite imagery range. In other words, the range of images must not be too sparse, and the relation to the land-use must have a similar pattern. However, with the ability of the model to allow the modification of the iteration loop, this allows the model to predict the expansion space of the buildings for more than 2 years. However, this research focuses on increasing the loop so that the model is able to predict future built-up areas to a satisfactory $R^2$ level. Table 1 displays the result of the matrix result changes. The comparison of land-use classification with visual interpretation data is shown in Tables A2–A4. From Appendix A the result of the error matrix for accuracy assessment of land-use types in 2014 found that the overall accuracy = 100% and with Kappa coefficient of agreement = 1.0.

**Table 1.** Result comparison of built-up growth simulation using the CA-Markov model with visual interpretation.

| Iterations-Loop | ($R^2$) from Image Simulated Using Transition Matrix 2014/2016 with Image Digitized in 2018 | ($R^2$) from Image Simulated Using Transition Matrix 2016/2018 with Image Digitized in 2018 |
|:---:|:---:|:---:|
| 1 | 0.91 | 0.95 |
| 2 | 0.87 | 0.92 |
| 3 | 0.85 | 0.91 |
| 4 | 0.81 | 0.86 |
| 5 | 0.77 | 0.79 |

In addition to the error matrix for accuracy assessment inspection in this study, it is used to create buffer areas out of the predict built-up to compare them with built-up from visual interpretation. The size of these two building areas is superimposed with the Overlay process.

Recursive loop assignments are defined only around the 1–5 range for a matrix to change the pitch of a 2-year interval to be used to synthesize the most appropriate iteration loop that is able to predict the expanding position of the built-up area in the future year, and also the precise test result of the model. Comparison of the positioning accuracy of the simulated image from the model to the size of the closed-space image translates from the satellite to the extent that the allowable configuration of the surplus boundary must be less than 0.04 sq. km, which indicates that the polygon is the same position.

The analysis result of Table 1 showed that the model could create a transition matrix from the years 2014–2016 to predict the position of the built-up polygon in the year 2018 and found that the $R^2$ is greater than the level 0.81 from the determination of iterations from the range of 1–4. The precision trend is gradually reduced when a loop is set to cycle 5. The accuracy trend is reduced gradually when a loop is set to five rounds and this indicates that the ability of the loop to be determined should not exceed four rounds to be used for future built-up position forecasts. Therefore, this research was made to predict the expansion position of the built-up area in the year 2022, which was constructed from the 2018 image range matrix, combined with four rounds of iterations. It is confirmed that the CA model can predict the buildings in both error matrix and buffer zone tests, with the confidence level of decision-making more than 81%.

The rise of built-up forecasts found that the area near the end of the Motorway-6 (yellow polyline) within a radius of 2 km has a higher growth rate than other areas. Although this area has a lower elevation than the other zone areas, it is also near the Lamtaklong River, as shown in Figure 3. Changes to the location of the built-up areas are mainly changes to the miscellaneous area. The water and road areas are rarely altered, although when more iterations are prescribed, the growth rate of the built-up area is higher, but the growth rate is relatively fixed within a radius of 5 km around the river. The result of the built-up area's expansion forecast in the year 2022 found that the built-up area had expanded along both sites of the road, because this zone area has a dense number of buildings and was evenly dispersed before the construction of the Highway was approved. The existing built-up area influences the model to calculate the transition matrix value, making the forecast with a very high-definition a highly-reproducible cycle, often with high accuracy. A cluster of existing buildings can also be attached, allowing the open space to be transformed into a built-up area with the driving force of economic and social factors, making fast changes to land utilization.

The application of a CA-Markov model in predicting the growth of most buildings considers the following points. The integration of the CA-Markov model is considered to be valuable for modelling land-use changes and is able to simulate and predict changes [42,43]. The CA-Markov model is the combination of Cellular Automata and transition probability matrix generated by the cross tabulation of two different images [42]. This combination of the CA-Markov model provides a robust approach in spatio-temporal dynamic modelling [42,44]. Furthermore, CA uses Markov to add spatial character to the model. On other words, the CA-Markov chain can simulate two-way transitions among any number of categories and can predict any transition among any number of categories [45,46]. It is worth mentioning that, the Cellular Automata is a dynamic process model that is used for the land-use cover change. This kind of model is common in the land-use modelling literature. Each cell with their own characteristics can represent parcels of land and can represent self-growth interactions as they are dynamic and reduplicate [47]. Furthermore, the land-use changes for any location (cells) can be clarified by the existing state and changes in neighbouring cells and can simulate the growth of things in two directions. This model is broadly used in spatial model for predicting future land-use [43,46].

The precision of the CA-Markov model needs to be monitored in use, but in this study it was confirmed that the model can be predicted at a satisfactory significant level. The CA-Markov model can predict the borderless growth of built-up in the Korat area. However, the change in land-use is a force that promotes built-up with only an expanding area zone, requiring high spatial resolution data to be used to construct replicas. The resolution of spatial data used in this study confirms that it is appropriate and directly affects the accuracy of the model. Sometimes, the precision of the model deviates with the position of the point in comparison with the result of the model. The accuracy test

in this study was used to randomly sample points of each type of land-use. Dividing a basin into a sub-size allows the CA-Markov model to generate a number of data to calculate the probability more precisely. Using a 30 m spatial resolution data can create a boundary that can lead to an independent variable value.

### 3.2. DEM Change and Hydrological Characteristics in 2022

The result of DEM modeling was used to predict the future of numerical height with a linear regression model. The data used for modeling is obtained from the 58 field observation points as a survey by setting the model's altitude measuring surveyor's telescope. The original altitude of the 2014 base year was from the Royal Thai Survey Department (RTSD) under the Ministry of Defense (MOD), Bangkok of Thailand [37]. The altitude information used as the basis for this study was made by flying with an aircraft of the RTSD agencies, so the spatial resolution is more detailed than other types of data available. It also updates this information up to date from exploring the field's altitude. This is in addition to the use of the proximity measurement method from the angle of contact with the river line with the function (Near) of the QGIS 3.6.0 to measure the distance between the altitude and the angle of contact with the river line. These factors are taken as independent variant information in a linear model. As a result, the data used to create the model in Table 2 results in Equation (14) and creates a data layer. The height of DEM in 2022 for hydrological characteristic analysis shows the altitude change of DEM, as Figure 4a–c.

$$DEM_{n+1} = 1.02\,(DEM_n) - (0.0000927 \times L_i) \tag{14}$$

where $DEM_{n+1}$ is ($Y$, dependent variable) the elevated point at any $i$ in 2022, $DEM_n$ is ($X1$) the elevated point of cell any $i$ in 2018 (meters) and $L_i$ is ($X2$) the distance between the center point of built-up polygons of feature any $i$ to the closet Lamtaklong stream line feature $s$ (meters), as shown in the data input of the model in Table 2. The DEM data is re-adjusted with the Fill tool and Sink to smooth surface adjustment and water flow.

A model with an $R^2$ value of 0.84 variable coefficient ($X2$) has a negative relationship with the altitude value. The utilization form of land buildings near the river is less than 2 km, and there is a higher filling rate than the areas further away. This linear model confirms that the built-up area is far away from the river, Most of these areas are the height of the area, less than the area near the river. Some areas have an increase in altitude, as these areas comprise the built-up type of housing, which must be filled at a height of more than 4 m.

The variable that influences altitude the most is DEM ($X1$). The elevation value of the $n$th base-year that is known to be influenced by the 1.02 model is when other independent variables have a value of 0. This linear regression model is used to predict the future height in the same position found to change the utilization of land to the built-up area. This cell is recalculated to the new level (1.02 × 180 = 183.60 m) which causes the altitude value of the elevation to be changed. The DEM model predicts the elevation map in 2022 from the use of image maps of built-up areas in 2022, imported, and processed in conjunction with the DEM-year-2018 old map, which has a height point updated from field exploration. This is shown in Table 2 and Figure 4.

Table 2. Data for DEM modeling.

| No. | Y (m) | $X_1$ (m) | $X_2$ (m) | Predicted (m) | Residual (m) | No. | Y (m) | $X_1$ (m) | $X_2$ (m) | Predicted (m) | Residual (m) |
|-----|-------|-----------|-----------|---------------|--------------|-----|-------|-----------|-----------|---------------|--------------|
| 1 | 165 | 163 | 3700 | 165.89 | 0.89 | 30 | 181 | 179 | 1300 | 180.38 | −0.62 |
| 2 | 167 | 164 | 3600 | 168.05 | 1.05 | 31 | 180 | 177 | 1100 | 179.22 | −0.78 |
| 3 | 172 | 167 | 2550 | 172.33 | 0.33 | 32 | 174 | 172 | 1800 | 173.99 | −0.01 |
| 4 | 180 | 175 | 1220 | 180.49 | 0.49 | 33 | 172 | 171 | 4000 | 172.38 | −0.38 |
| 5 | 187 | 183 | 1201 | 186.83 | −0.17 | 34 | 180 | 179 | 1500 | 180.54 | 0.54 |
| 6 | 165 | 160 | 3405 | 164.63 | −0.37 | 35 | 187 | 184 | 900 | 185.82 | −1.18 |
| 7 | 192 | 187 | 550 | 191.99 | −0.01 | 36 | 165 | 163 | 4200 | 163.65 | −1.35 |
| 8 | 176 | 173 | 4450 | 176.58 | 0.58 | 37 | 192 | 188 | 600 | 191.01 | −0.99 |
| 9 | 188 | 184 | 375 | 186.23 | −1.77 | 38 | 176 | 173 | 4200 | 176.55 | 0.55 |
| 10 | 179 | 176 | 2500 | 178.50 | −0.50 | 39 | 188 | 185 | 980 | 187.27 | −0.73 |
| 11 | 186 | 183 | 1112 | 184.49 | −1.51 | 40 | 179 | 174 | 1700 | 177.53 | −1.47 |
| 12 | 185 | 181 | 800 | 184.36 | −0.64 | 41 | 186 | 184 | 960 | 187.50 | 1.50 |
| 13 | 182 | 176 | 1050 | 180.40 | −1.60 | 42 | 185 | 184 | 1300 | 184.41 | −0.59 |
| 14 | 181 | 177 | 900 | 180.40 | −0.60 | 43 | 181 | 179 | 1200 | 180.38 | −0.62 |
| 15 | 180 | 177 | 920 | 178.05 | −1.95 | 44 | 180 | 177 | 1100 | 179.12 | −0.88 |
| 16 | 174 | 170 | 3200 | 174.23 | 0.23 | 45 | 174 | 171 | 2560 | 174.15 | 0.15 |
| 17 | 177 | 173 | 1200 | 177.16 | 0.16 | 46 | 177 | 173 | 2700 | 178.27 | 1.27 |
| 18 | 179 | 176 | 800 | 180.39 | 1.39 | 47 | 179 | 176 | 2100 | 179.41 | 0.41 |
| 19 | 182 | 178 | 920 | 182.49 | 0.49 | 48 | 182 | 180 | 1400 | 183.55 | 1.55 |
| 20 | 187 | 181 | 945 | 186.43 | −0.57 | 49 | 187 | 185 | 960 | 186.38 | −0.62 |
| 21 | 180 | 179 | 500 | 181.45 | 1.45 | 50 | 180 | 178 | 1300 | 179.35 | −0.65 |
| 22 | 181 | 180 | 3600 | 180.65 | −0.35 | 51 | 181 | 176 | 1250 | 182.65 | 1.65 |
| 23 | 192 | 188 | 1300 | 191.09 | −0.91 | 52 | 192 | 189 | 700 | 191.12 | −0.88 |

**Table 2.** *Cont.*

| No. | Y (m) | $X_1$ (m) | $X_2$ (m) | Predicted (m) | Residual (m) | No. | Y (m) | $X_1$ (m) | $X_2$ (m) | Predicted (m) | Residual (m) |
|---|---|---|---|---|---|---|---|---|---|---|---|
| 24 | 176 | 174 | 1700 | 174.52 | −1.48 | 53 | 176 | 174 | 3200 | 177.57 | 1.57 |
| 25 | 188 | 185 | 1300 | 186.31 | −1.69 | 54 | 188 | 186 | 980 | 188.33 | 0.33 |
| 26 | 179 | 176 | 1700 | 177.31 | −1.69 | 55 | 179 | 177 | 1700 | 179.30 | 0.30 |
| 27 | 186 | 174 | 1300 | 184.45 | −1.55 | 56 | 180 | 177 | 1900 | 180.35 | 0.35 |
| 28 | 185 | 181 | 1250 | 183.38 | −1.62 | 57 | 181 | 177 | 1400 | 190.95 | −0.05 |
| 29 | 182 | 177 | 1100 | 182.40 | −0.40 | 58 | 175 | 171 | 4520 | 174.40 | −0.60 |

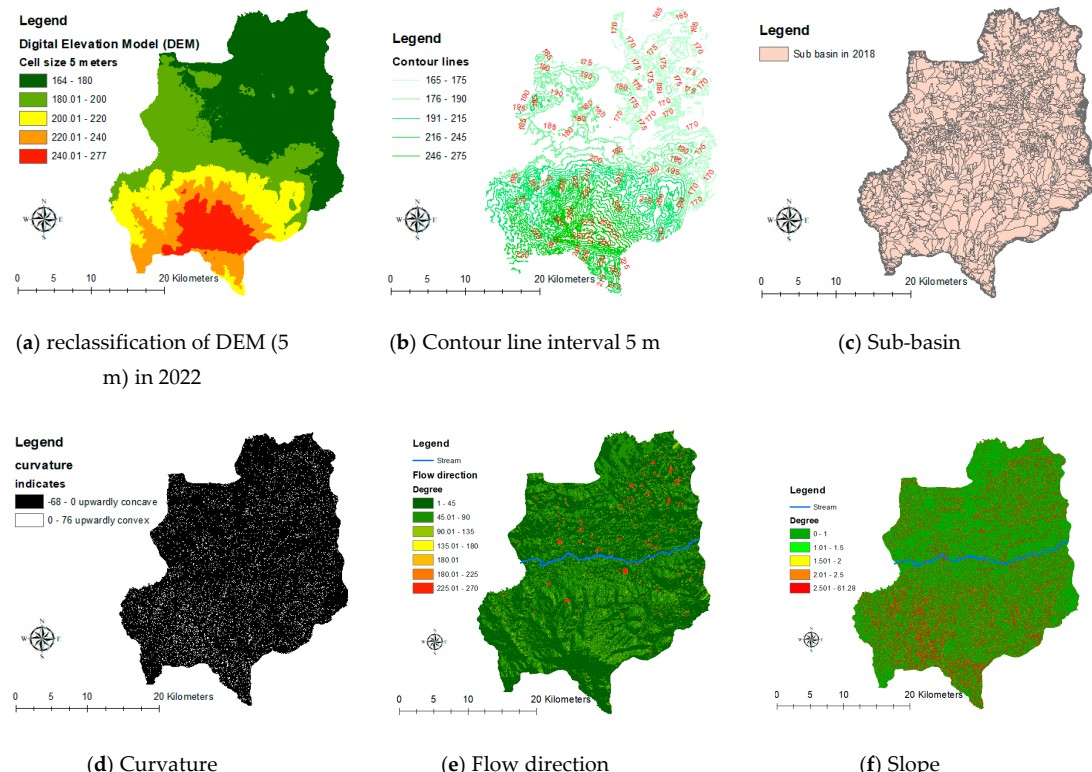

**Figure 4.** Maps of hydrological data (**a**–**f**) to use for independent variables extraction of FRS-GWR model in 2022.

From Figure 5a–c, the average altitude is seen to have significantly changed in the upper area of the Lamtaklong River mainly, and the result of significant hydrologic change in Table 3, the number of sub-basins generated from the data DEM Years 2014, 2018, and 2022, found that there was a total number of 2325, 2151, 2050 polygons, respectively. The number of sub-basins tends to be less likely to decline in future years as they increase the level of numerical height in the joint's area of the sub-basin region. The area of any adjacent sub-basin is fused together and water can flow to each area within the basin. This results in a longer flow sequence length from 878.47, 888.58, and 890.32 km in 2014, 2018, and 2022 respectively. There are several simple studies that say that DEM is converted to affect the nature of the basin area. The definitions of drainage and relief are essential for understanding spatial differences within the catchment [48]. Drainage density has been found to correlate with valley density, channel head source area, relief, climate, vegetation, soil and rock properties, and landscape evolution processes [49]. Measuring drainage density is extremely difficult, and it relies on good topographic maps at a detailed scale [50,51]. As an alternative to drainage density, the parameter of potential drainage density is often obtained from a digital elevation model (DEM) [52]. Drainage density has been found to correlate with valley density, channel head source area, relief, climate, vegetation, soil and rock properties, and landscape evolution processes [49]. Analysis by Pal and Saha [48] showed a high correlation between drainage density and the following parameters: length of overland flow, number of stream junctions in the basin, and the infiltration coefficient and drainage texture [48]. Low drainage diversity was related to low drainage density. A guideline for modeling of this research synthesized variables that influence the changing of the height in a physical area into an independent variant, which allows the modeling to predict the hydrologic change.

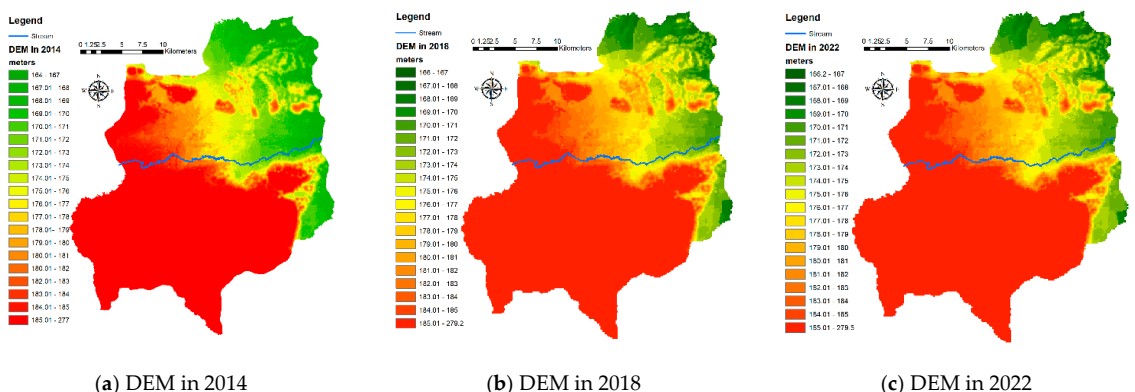

(**a**) DEM in 2014         (**b**) DEM in 2018         (**c**) DEM in 2022

**Figure 5.** Maps of DEM in (**a**) Year 2014, (**b**) Year 2018 and (**c**) Year 2022.

**Table 3.** DEM change impact to basic hydrological characteristics.

| DEM of Year | Number of Sub-Basins | Number of Sub-Basins in Upper Streams | Length of Stream Order 3 to 6 (Kilometers) | Average Elevation Range (165–175 m) in Upper Streams |
|---|---|---|---|---|
| 2014 | 2325 | 1185 | 878.47 | 173.54 |
| 2018 | 2151 | 1078 | 888.58 | 175.45 |
| 2022 | 2050 | 984 | 890.32 | 177.58 |

The hydrologic base data that is used to calculate the index of independent variables of the year 2022. Figure 4a is a 5 m data class DEM, a resolution of 5 m, which is divided into a range of 180 over 20 m in height. The DEM 2022 data is used as the basis for the creation of all other information; (b) contour line interval 5 m; (c) any sub-basin; (d) surface curvature; (e) flow direction; and (f) slope.

The entire data layer uses the edited data modification process (manipulation using GIS process) to be able to analyze the Geographic Information System (GIS) by assigning the cell sizes of the raster data to 5 m. The data processing takes a long time, but the results show the resolution and amount of descriptive data that will improve the accuracy of the GWR model.

The map in Figure 6 displays that the stream order is used to level 3 to 6, because it is a level with the embodiment of the flow outlet with the flow point from high to low. The stream order generated from DEM in 2018 compared to the year 2022 found that the stream order in 2022 is longer than that of the year of 2018 in every buffer distance. The rise of the building position affects the direction of the water flow. This is observed by comparing the stream length and stream order along the buffer distance every 1000 m out of the Lamtaklong River, as shown in Figure 7 and Table 4.

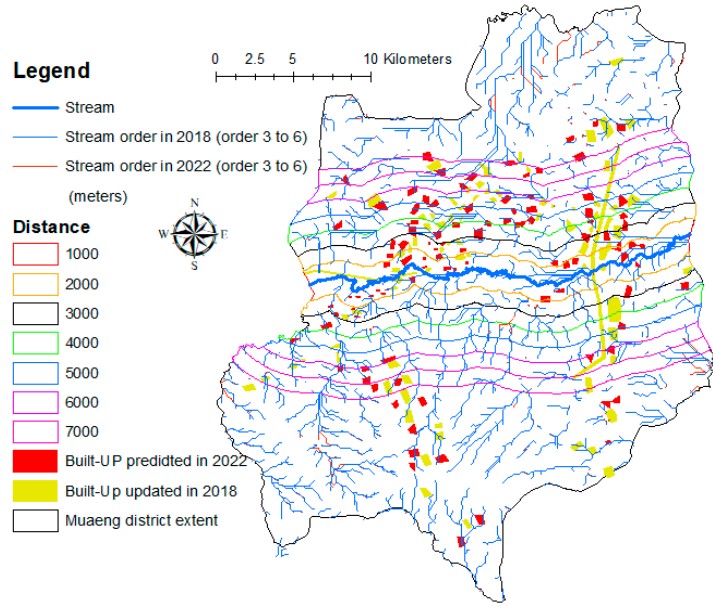

**Figure 6.** Map of stream order 3 to 6 with buffering from Lamtaklong sreamline.

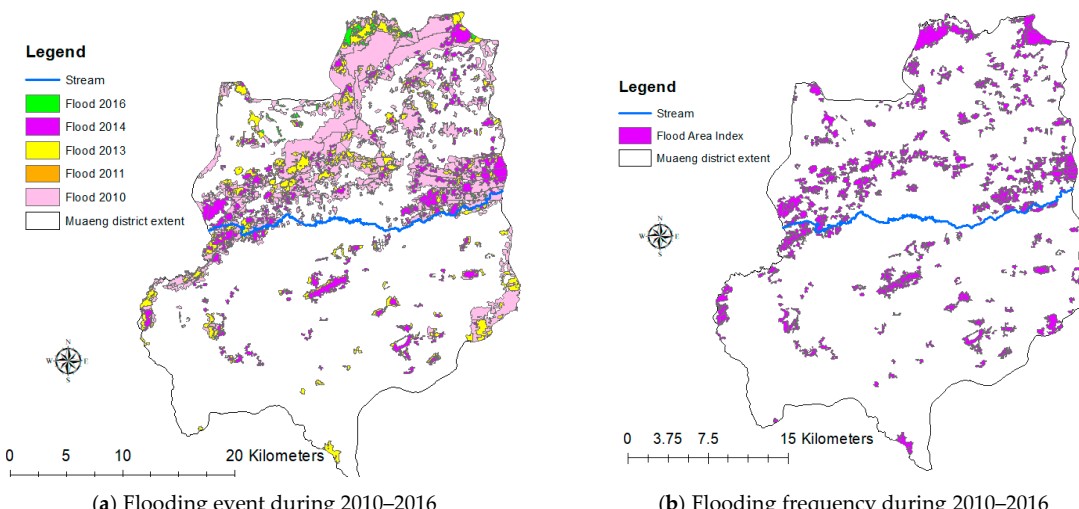

(**a**) Flooding event during 2010–2016　　　　　　　(**b**) Flooding frequency during 2010–2016

**Figure 7.** Maps of flood change in (**a**) and Flood Area Index (*FAI*) in (**b**).

**Table 4.** Comparison of stream order length of Year 2018/2022.

| Distance from Lamtaklong Stream (Meters) | Stream Order in 2018 | Stream Order in 2022 |
|:---:|:---:|:---:|
| | Order 3 to 6 (Length KM.) | Order 3 to 6 (Length KM.) |
| 1000 | 56.7 | 56.5 |
| 2000 | 63.9 | 71.2 |
| 3000 | 73.6 | 75.4 |
| 4000 | 85.4 | 86.7 |
| 5000 | 86.1 | 87.3 |
| 6000 | 87.4 | 88.7 |
| 7000 | 87.6 | 88.9 |

The longer stream length takes much longer to drain. In a buffer space of up to 1000 m, most of them have a slight slope, allowing the analysis of the length of the flow sequence to be less than the

buffer range area away from the river. Also, in the buffer range, 2000–3000 m toward the west is found to have a very dense flow sequence of water.

The area has many large road construction projects. This causes changes to the route, water flow, and creates a lack of flow lines. This area is frequently affected by flooding. Data from the DEM model contributes to the *FRS-GWR* modeling support, as well as the discussion of the results.

### 3.3. Flood Area Index (FAI)

The FAI index shows the severity of the area where flooding is most frequent from the use of flood information over a 5-year period. The retrospective of the flood area in the distance edge of about 5 km from the Lamtaklong stream line is where the FAI index is taken. The link is the spatial database (Attribute) of a variable by ($Y$) to the data layer. The sub-boundary of the year 2018 is used to create the GWR model together with other independent variables.

The Dependent variables in this study were used to create an FAI index to find the frequency of floods in the area with the lead data layer. The flood boundaries are converted into binary variables (binary). If the area that has flooded is set to 1, and otherwise is 0, the analysis process requires the conversion of the vector layer (.SHP) to raster (.TIFF) and is used in overlay function with the binary model using the raster calculation function in QGIS version 3.6.0.

The high values of the flood area index are scattered in the upper area of the study area (depicted in Figure 7) and the position of the flood reaches an average altitude of about 164–167 m, while the flood area has the characteristics of land-use as agriculture; this area has also experienced flooding from the city center, making the flood boundary overlap with other areas. The average of DEM in the range of 165 to 175 m is consistent with the analysis results from FAR, thus providing a high index in the same area.

### 3.4. GWR Model and Flood Risk Susceptible (FRS) in 2022

Table 5 summarizes the results of the Global and GWR models of study areas in the Muaeng district of Nakhon Ratchasima Province. The Monte Carlo Test summary table and the GWR model calibration found that five out of the eight significant independent variables show spatial non- stationarity. In addition, the GWR model has an $R^2$ level precision at 85%, which is greater than the global model ($R^2 = 0.57$), the GWR replica creates a layer of free variable data in the GIS data layer with the interpolation method to show the spatial variation, as shown as the map in Figure 8b–f. An $R^2$ value of GWR local operation which is significant to the forecast has a value in the range of 0.8–0.88 and is an average, thus giving 0.6, and the global result of the $R^2$ value is 0.57. It can be assumed that the relationship between the selected factors and in the created city will be captured by the GWR model in those regions [41]. The growth of built-up area use in the region with low $R^2$ may be affected to a greater degree by other factors that were not considered in this study, and there may also be fringe effects that were also not considered. In the local operation of GWR, an F-test was also used to test whether spatial changes exist in the relationship under the study [41], specifically testing whether the GWR model is updated and explains significantly the relationship over the general global operation using Ordinary Least Square (OLS).

**Table 5.** Summary results of the flood risk susceptibility FRS-GWR model.

| Flood Risk Susceptibility GWR Model Coefficients | | | |
|---|---|---|---|
| | **GLOBAL GWR** | | |
| **Variables** | $\beta$ | $t$ | $p$-Value [a] |
| Intercept | 40.56 | 4.16 *** | 0.00 *** |
| Normalized Digital Elevation Model Index (*NDEMI*, *X1*) | −26.03 | −1.61 *** | 0.01 *** |
| Normalized Contour Index (*CLI*, *X2*) | −21.00 | −3.87 *** | 0.84 *n/s* |
| Area of Built-up Index (*ABUI*, *X3*) | 1.35 | 5.25 *** | 0.00 *** |
| Density of Built-up ($D_j$, *X4*) | 4.82 | 6.58 *** | 0.00 *** |
| Curvature Index (*CI*, *X5*) | −0.57 | −3.10 *** | 0.00 *** |
| Slope length Index (*SLI*, *X6*) | −0.48 | −2.17 *** | 0.00 *** |
| Perimeters ($P_j$, *X7*) | 0.03 | 0.29 *n/s* | 0.95 *n/s* |
| Sub-basin Area ($S_j$, *X8*) | 0.05 | 0.38 *n/s* | 0.99 *n/s* |
| *N* | 2151 | | |
| Adjusted R$^2$ | 0.57 | | 0.88 |

*** = significant at 1% level. *n/s* = not significant. [a] Results of Monte Carlo test for spatial non-stationarity [43,44].

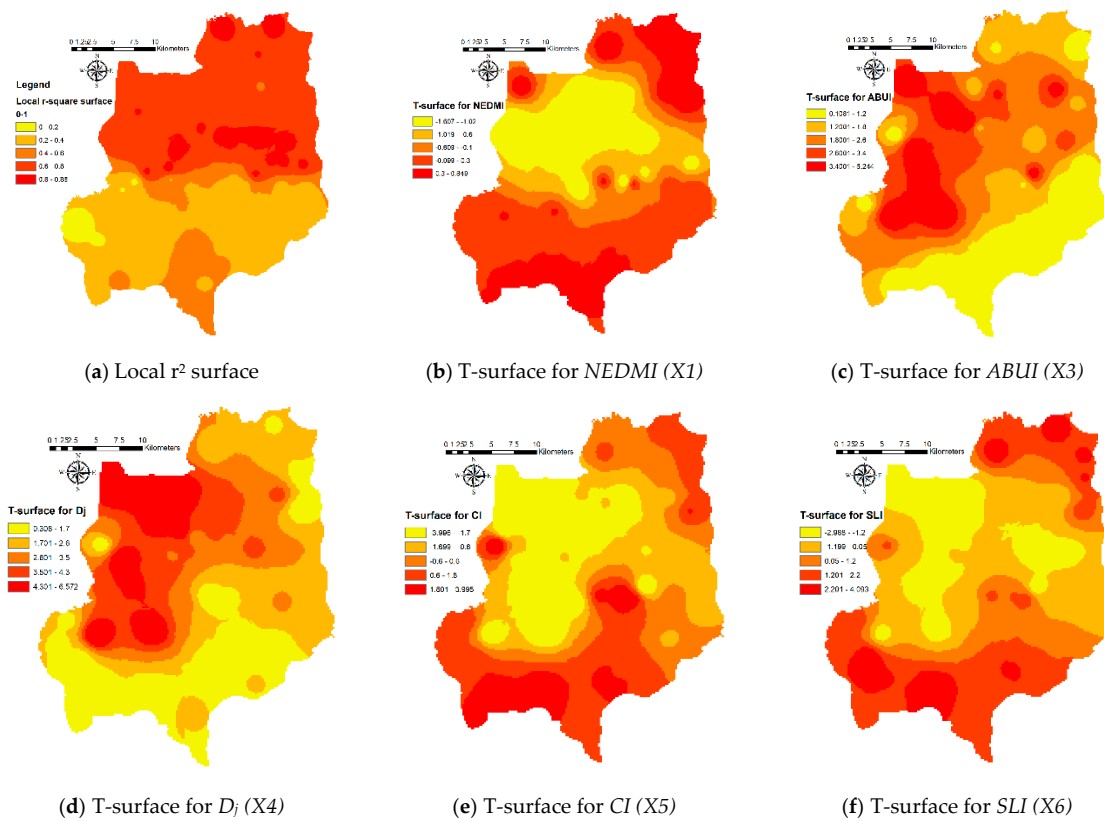

(**a**) Local r² surface　　　　(**b**) T-surface for *NEDMI (X1)*　　　　(**c**) T-surface for *ABUI (X3)*

(**d**) T-surface for *$D_j$ (X4)*　　　　(**e**) T-surface for *CI (X5)*　　　　(**f**) T-surface for *SLI (X6)*

**Figure 8.** Maps of Local r-square (**a**) and T surface distribution of FRS-GWR model (**b**–**f**).

The analysis of variance was addressed through the testing of ANOVA for the creation in the city of Korat. The F-value was 8.231. The high F-value suggests that the GWR format was significantly improved through a global form to define the relationship between the built-up growth and different factors. Additionally, the Akaike information (AIC) of the GWR format (178,474.5) is far less than one of the global operations (20,371.1). This indicates that the GWR local operation is improved more than the OLS model (referenced in Table 6).

**Table 6.** ANOVA test of the FRS-GWR over the OLS regression model.

| Source | SS | DF | MS | F |
|---|---|---|---|---|
| OLS Residuals | 56,312.3 | 26.0 | | |
| GWR Improvement | 27,006.7 | 465.36 | 4.72 | |
| GWR Residuals | 29,305.6 | 3210.53 | 7.84 | 8.231 |
| GWR Akaike Information Criterion | 178,474.5 | (OLS): 20,371.1 | | |

The *FRS-GWR* model predicts the area risk of flood sensitivity in the year 2022. The significant independent variables were only selected containing the normalized digital elevation model Index (*NDEMI*, *X*1), area of built-up index (*ABUI*, *X*3), density of built-up (*D$_j$*, *X*4), curvature index (*CI*, *X*5), and slope length Index (*SLI*, *X*6) and to create a model *FRS-GWR* as Equation (15) when assigning *Y* values from GWR model = *F$_j$* in Equation (5). The result of the dispersion of independent variables is as follows Figure 8b–f.

$$FRS_{2022} = (4.16 - (1.61NEDMI) + (5.25ABUI) + (6.58D_j) - (3.1CI) - (2.17SLI))x100/S_j \qquad (15)$$

The independent variables are selected from the relationship with the variable as a significant factor. The surface distribution coefficient (T-surface) (depicted in Figure 8) that is consistent in the independent variable area *NEDMI (X1)* shows the dispersion of negative range coefficients; in addition, however, the higher the R$^2$ high value, the higher the flooding risk. In accordance with the effect of *CI (X5)* and *SLI (X6)*, but contrary to the positive coefficient of *ABUI (X3)* and *D$_j$ (X4,)* is when there is space and the density of more built-up areas will hinder the flow of water. Additionally, the variable that is not mentioned does not have a compliance with *Y*.

The risk area forecast for flood sensitivity *FRS-GWR* model in 2022 uses the filtered independent variant data that is significant and influences the variables followed by the GIS tools simulated from a hydrologic analysis and then imported into Equation (15). Figure 9 displays the flood risk susceptibility values that range from 0.6 to 0.86, and this highlights the potential for flooding in the area when it rains. For approximately 60–86 percent of the space, most of the area is more than 70 percent of the upper basin above the Lamtaklong River. There are index values ranging from 50, but there are some areas with a lower flooding potential.

For areas with DEM ranging from 180 to 200 m in the lower part of the sub-basin, most of which is less prone to flooding as the area has an average altitude of more than the other zones, except the Eastern zone area, the index value is 40–45 because it is close to the Lamtaklong River. As well as in the year 2022, expanding the construction of the buildings in this area is rapidly emerging from the new road-building influence in the year 2018.

An *FRS-GWR* model compared to (FAI) was found to be consistent, but the *FRS-GWR* map has a more continuous dispersion of the area than the *FRS-GWR* model can predict. The sensitive area in terms of the risk of flooding and the ideal analysis in the area is not very large, and there is a large number of sub-areas (spatial units) and the continuous dispersion will greatly improve the forecast performance.

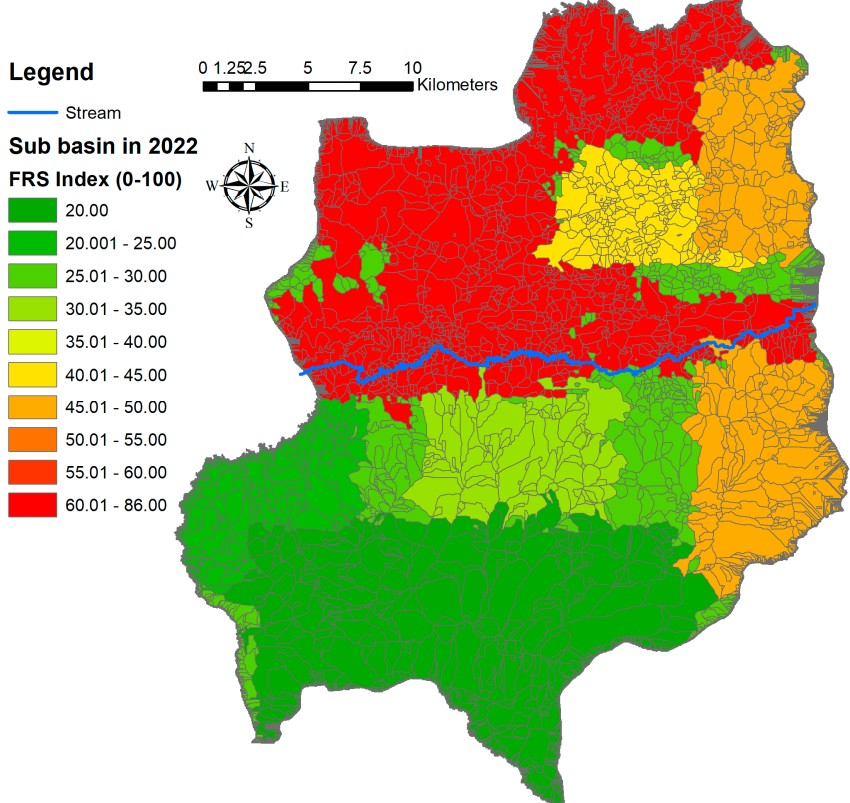

**Figure 9.** Map of flood risk susceptible forecast using *FRS*-GWR model in 2022.

## 4. Conclusions

A future building forecast with a CA-Markov model in this study found that it can be adjusted to require repeated process assignments (Iterations), making it possible to use images to predict future buildings for many years to come. This study uses land utilization data every 2 years to predict the position of buildings that are arising up to 4 years, using 2014/2016 data to predict the building of the year 2018. ($R^2$ value is also greater than 0.8). In general, predictive research with a CA-Markov model can be used to create a map based on data spacing. According to Bertaud [53], it is important to note that the areas that are configured in this way require a dispersion of the land cover that requires continuous dispersion.

This research also found that the GWR modeling provides precise prediction capabilities. It is necessary to create a layer of spatial units to be appropriate for use. Reference by Brunsdon [54], explains how the unit of this sub-area is different depending on the spatial relationship of independent variables and the variables based on this research. This research hypothesized that the data layer DEM with a higher average level will affect other hydrologic data changes and cause the sub-watershed boundary (sub-basin) to change. In order to be used to extract variable data, as well as independent variables, to be consistent and relevant in the testing area, various statistical values, including local $R^2$, AIC, and F-test, were used to confirm that the GWR model is more accurate than the GLOBAL model (OLS) in every test. The research developed the GWR model in conjunction with a simulated flood frequency index model to analyze risk areas of flood sensitivity (*FRS-GWR*) as confirmed by a model that can predict the sub-basin that has potential to be at risk of flooding in future years from a comparative test against flood data from the past.

Modeling guidelines for the relationship of spatial heights affect the analysis of water flow in both the basin region, flow accumulation, flow direction and stream order. Reference by Jenkins [29], stated that if there are many adjacent built-ups, there will result a change from an ever-changing area to a water-inlet area because the water has no outlet.

However, further research in other areas may be added to the use of TIN data (Tri Irregular Network) in conjunction with the analysis of the flow of water to provide a more complete flow direction, as well as analysis of the accumulation of water (flow accumulation). Analysis of flow accumulation enables a more continuous flow of water to accumulate and this can be used to drain the water out of the sub-basin area. However, analysis with this information is only appropriate to work in small spaces, as this takes longer to process than the data DEM.

This study can be used as a prototype to analyze environmental impacts, especially the flow and barriers of water flow from the further expansions of buildings in other areas. Reference [40] explains the modeling research in the future to assess the flood-prone area to be included in plans in order to better cope with climate change that affects the frequency and risk of flood disasters. In addition, this research will allow stakeholders to better plan their development of urban areas so as to minimize environmental impacts and make plans for the sustainable growth of a city.

**Author Contributions:** Conceptualization, P.L. and N.B.; Formal analysis, P.L. and N.B.; Investigation, N.B.; Methodology, P.L.; Project P.L and N.B.; Writing—original draft, P.L. and N.B.; and Writing—review and editing, P.L. and N.B.

**Funding:** This research project is financially supported by Faculty of Informatics, Mahasarakham University (Fast Track 2019) and Research unit of Geo-informatics for Local Development and Climate Changes, Mitigation and Adaptation Research Unit; CMARE).

**Conflicts of Interest:** The authors declare no conflict of interest.

## Appendix A

**Table A1.** Allocation for land-use categories in 2014, 2016, and 2018.

| Land-Use Types | 2014 | | 2016 | | 2018 | |
|---|---|---|---|---|---|---|
| | sq. km | % | sq. km | % | sq. km | % |
| Urban and built-up area (U) | 202.68 | 31.40 | 209.45 | 32.43 | 225.88 | 34.97 |
| Agriculture land (A) | 385.01 | 59.61 | 382.05 | 59.15 | 370.06 | 57.30 |
| Forest land (F) | 7.12 | 1.10 | 7.03 | 1.08 | 6.52 | 1.01 |
| Water body (W) | 7.45 | 1.15 | 7.06 | 1.09 | 6.91 | 1.07 |
| Miscellaneous land (M) | 43.58 | 3.33 | 40.25 | 6.25 | 36.47 | 5.65 |
| Total | 645.84 | 100.00 | 645.84 | 100.00 | 645.84 | 100.00 |

**Table A2.** Error matrix for accuracy assessment of land-use types in 2014.

| Land-Use Types in 2014 | Reference Data in 2018 | | | | | |
|---|---|---|---|---|---|---|
| | U | A | F | W | M | Total |
| Urban and built-up area (U) | 43 | 0 | 0 | 0 | 0 | 43 |
| Agriculture land (A) | 0 | 58 | 0 | 0 | 0 | 58 |
| Forest land (F) | 0 | 0 | 3 | 0 | 0 | 3 |
| Water body (W) | 0 | 0 | 0 | 15 | 0 | 15 |
| Miscellaneous land (M) | 0 | 0 | 0 | 0 | 18 | 18 |
| Total | 43 | 58 | 3 | 15 | 18 | 137 |

Note: 1. Overall accuracy = 100 %. 2. Kappa hat coefficient of agreement = 1.0.

**Table A3.** Error matrix for accuracy assessment of land-use types in 2016.

| Land-Use Types in 2016 | Reference Data in 2018 | | | | | |
|---|---|---|---|---|---|---|
| | U | A | F | W | M | Total |
| Urban and built-up area (U) | 57 | 0 | 0 | 0 | 0 | 57 |
| Agriculture land (A) | 0 | 60 | 0 | 0 | 0 | 60 |
| Forest land (F) | 0 | 0 | 2 | 0 | 0 | 2 |
| Water body (W) | 0 | 0 | 0 | 14 | 0 | 14 |
| Miscellaneous land (M) | 0 | 0 | 0 | 0 | 16 | 16 |
| Total | 57 | 60 | 2 | 14 | 16 | 149 |

Note: 1. Overall accuracy = 100 %. 2. Kappa hat coefficient of agreement = 1.0.

**Table A4.** Error matrix for accuracy assessment of land-use types in 2018.

| Land-Use Types in 2018 | Reference Data in 2018 | | | | | |
|---|---|---|---|---|---|---|
| | U | A | F | W | M | Total |
| Urban and built-up area (U) | 65 | 0 | 0 | 0 | 0 | 65 |
| Agriculture land (A) | 0 | 63 | 0 | 0 | 0 | 63 |
| Forest land (F) | 0 | 0 | 2 | 0 | 0 | 2 |
| Water body (W) | 0 | 0 | 0 | 12 | 0 | 12 |
| Miscellaneous land (M) | 0 | 0 | 0 | 0 | 14 | 14 |
| Total | 65 | 63 | 2 | 12 | 14 | 156 |

Note: 1. Overall accuracy = 100 %. 2. Kappa hat coefficient of agreement = 1.0.

**Table A5.** Transition area matrix for land-use change between 2014 and 2016.

| Land-Use in 2016 | Land-Use Types (sq. km) | | | | | |
|---|---|---|---|---|---|---|
| | U | A | F | W | M | Total |
| Urban and built-up area (U) | 209.45 | 0.00 | 0.00 | 0.00 | 0.00 | 209.45 |
| Agriculture land (A) | 31.52 | 330.75 | 0.00 | 1.15 | 18.63 | 382.05 |
| Forest land (F) | 0.05 | 0.62 | 5.21 | 0.01 | 1.14 | 7.03 |
| Water body (W) | 0.03 | 0.56 | 0.00 | 6.01 | 0.46 | 7.06 |
| Miscellaneous land (M) | 15.34 | 1.02 | 0.00 | 0.21 | 23.68 | 40.25 |
| Total | 256.39 | 332.95 | 5.21 | 7.38 | 43.91 | 645.84 |

**Table A6.** Transition probability matrix for land-use change between 2014 and 2016.

| Land-Use in 2016 | Land-Use Types | | | | | |
|---|---|---|---|---|---|---|
| | U | A | F | W | M | Total |
| Urban and built-up area (U) | 1.000 | 0.000 | 0.000 | 0.000 | 0.000 | 1.000 |
| Agriculture land (A) | 0.083 | 0.866 | 0.000 | 0.003 | 0.048 | 1.000 |
| Forest land (F) | 0.008 | 0.088 | 0.741 | 0.001 | 0.162 | 1.000 |
| Water body (W) | 0.004 | 0.079 | 0.000 | 0.852 | 0.065 | 1.000 |
| Miscellaneous land (M) | 0.381 | 0.025 | 0.000 | 0.005 | 0.589 | 1.000 |

**Table A7.** Transition area matrix for land-use change between 2016 and 2018.

| Land-Use in 2018 | Land-Use Types (sq. km) | | | | | |
|---|---|---|---|---|---|---|
| | **U** | **A** | **F** | **W** | **M** | **Total** |
| Urban and built-up area (U) | 225.88 | 0.00 | 0.00 | 0.00 | 0.00 | 225.88 |
| Agriculture land (A) | 35.61 | 316.87 | 0.00 | 1.02 | 16.56 | 370.06 |
| Forest land (F) | 0.00 | 0.23 | 5.15 | 0.02 | 1.12 | 6.52 |
| Water body (W) | 1.23 | 0.03 | 0.00 | 4.01 | 1.64 | 6.91 |
| Miscellaneous land (M) | 14.56 | 0.95 | 0.00 | 0.14 | 20.82 | 36.47 |
| Total | 277.28 | 318.08 | 5.15 | 5.19 | 40.14 | 645.84 |

**Table A8.** Transition probability matrix for land-use change between 2016 and 2018.

| Land-Use in 2018 | Land-Use Types | | | | | |
|---|---|---|---|---|---|---|
| | **U** | **A** | **F** | **W** | **M** | **Total** |
| Urban and built-up area (U) | 1.000 | 0.000 | 0.000 | 0.000 | 0.000 | 1.000 |
| Agriculture land (A) | 0.096 | 0.856 | 0.000 | 0.003 | 0.045 | 1.000 |
| Forest land (F) | 0.000 | 0.035 | 0.789 | 0.003 | 0.173 | 1.000 |
| Water body (W) | 0.179 | 0.004 | 0.000 | 0.580 | 0.237 | 1.000 |
| Miscellaneous land (M) | 0.399 | 0.026 | 0.000 | 0.003 | 0.572 | 1.000 |

**Table A9.** Transition area matrix for land-use change between 2018 and 2022.

| Land-Use in 2022 | Land-Use Types (sq. km) | | | | | |
|---|---|---|---|---|---|---|
| | **U** | **A** | **F** | **W** | **M** | **Total** |
| Urban and built-up area (U) | 242.85 | 0.00 | 0.00 | 0.00 | 0.00 | 242.85 |
| Agriculture land (A) | 42.62 | 300.06 | 0.00 | 1.03 | 18.60 | 362.31 |
| Forest land (F) | 0.00 | 0.01 | 6.17 | 0.02 | 0.02 | 6.22 |
| Water body (W) | 1.58 | 0.82 | 0.00 | 2.73 | 1.84 | 6.97 |
| Miscellaneous land (M) | 15.64 | 0.20 | 0.00 | 0.18 | 11.47 | 27.49 |
| Total | 302.69 | 301.09 | 6.17 | 3.96 | 31.93 | 645.84 |

Iterations loop = 4.

**Table A10.** Transition probability matrix for land-use change between 1986 and 1994.

| Land-Use in 2022 | Land-Use Types | | | | | |
|---|---|---|---|---|---|---|
| | **U** | **A** | **F** | **W** | **M** | **Total** |
| Urban and built-up area (U) | 1.000 | 0.000 | 0.000 | 0.000 | 0.000 | 1.000 |
| Agriculture land (A) | 0.117 | 0.828 | 0.000 | 0.003 | 0.052 | 1.000 |
| Forest land (F) | 0.000 | 0.002 | 0.991 | 0.003 | 0.023 | 1.000 |
| Water body (W) | 0.227 | 0.118 | 0.000 | 0.392 | 0.263 | 1.000 |
| Miscellaneous land (M) | 0.569 | 0.007 | 0.000 | 0.007 | 0.417 | 1.000 |

**Table A11.** Data and Equipment.

| Data and Equipment | Date | Number of Date (sheet) | Scale | Source/Remarks |
|---|---|---|---|---|
| 1. RS/GIS Data Types | | | | |
| 1.1. primary datasets-Satellite imagery | December 2014/2016/2018 | 750,000 sheets | 0.9 m × 0.9 m | Google Earth Pro |
| -Topographic map (5438IV) | 2018 | 1 | 1:50,000 | Royal Thai Survey Department (RTSD) |
| 1.2. Secondary datasets | | | | |
| -Land-use | 2014/2016/2018 | - | 1:4,000 | Land Development Department (LDD) and updated from field observation |
| Road | 2018 | - | - | Department of Highways (DOH) and updated from field observation |
| Stream | 2016 | - | - | Department of Water Resources |
| Contour line | - | - | - | field observation |
| DEM | 2018 | - | 5 m × 5 m | (RTSD), FGDS from Geo-Informatics and Space Technology Department Agency (GISTDA) and updated from field observation |
| 2. Equipment hardware and software | | | | |
| 2.1. GPS | | | | -Research unit of Geo-informatics for Local Development |
| 2.2. Notebook (Acer Aspire VX15) | | | | -Personal |
| 2.3 Software - QGIS 3.6.0 - IDRISI 15.0 - ArcGIS 9.3.1 | | | | -Free -Personal -Research unit of Geo-informatics for Local Development |

**Table A12.** List of abbreviation.

| The Description | Abbreviation |
|---|---|
| Akaike Information Criterion | AIC |
| Area of Built-Up Index | (ABUI, X3) |
| Arc Geographic Information System | ArcGIS |
| Cellular Automata | (CA) |
| Conversion of land use changes and its Effects at Small regional extent | CLUE-S |
| Curvature Index | (CI, X5) |
| Degree of Freedom | DF |
| Density of Built-Up | ($D_j$, X4) |
| Digital Elevation Model | (DEM) |
| Flood Risk Susceptibility (FRS) in GWR | FRS-GWR |
| F-Ratio | F |
| Geographic Weighted Regression | (GWR) |

**Table A12.** *Cont.*

| The Description | Abbreviation |
| --- | --- |
| Geographic Resources Analysis Support System | GRASS |
| Integrated Geographic Information System | (GIS) |
| and remote sensing software | IDRISI |
| Land Transformation Model | LTM |
| Land use | (LU) |
| Mean square | MS |
| Nakhon Ratchasima province | some called as Korat |
| Normalized Contour Index | (CLI, X2) |
| Normalized Digital Elevation Model Index | (NDEMI, X1) |
| Ordinary Least Square | OLS |
| Quantum GIS software package | QGIS 3.6.0 |
| Raster format in IDRISI | (.RST) |
| Slope Length Index | (SLI, X6) |
| Sum of Square | SS |
| Vector format in ArcGIS | (.SHP) |

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
