# Peer review of "Built-Up Growth Impacts on Digital Elevation Model and Flood Risk Susceptibility Prediction in Muaeng District, Nakhon Ratchasima (Thailand)"

_water, doi:10.3390/w11071496_

Round 1

Reviewer 1 Report

Brief reviewer(s)' comments to author:

Manuscript Title: Spatial Modeling of Built-up Growth Prediction and Its Impact on Digital Elevation Model and Flood Risk Susceptibility in Muaeng District, Nakhon Ratchasima, (Thailand).

Reviewer summary of manuscript

The authors downloaded and used high-resolution multi-temporal remote sensing image data from IKONOS sensor (between 2014 and 2018) to predict built-up areas for 2022 using optimized Cellular Automata (CA) model. To derive the transition of digital elevation model (DEM), the authors used the linear regression modelling method. Then they analysed the risk of the stativity of flood areas in the study area by applying the Geographic Weighted Regression (GWR) approach.

Results confirmed plausibility of using optimized CA to generate accurate prediction of the expansion of built-up areas in 2022 using LU data within a 2-year interval. The authors reported the convergence of the predicting model at iteration no. 4, which included spatial locations and ground-water touch points of the construction and was used to estimate and model DEM to extract independent hydrology variables that will be used in the determination of Flood Risk Susceptibility (FRS).

Recommendation

This is an interesting paper because it explored the use of remote sensing data and mathematical approaches spatially model of built-up growth prediction to identify the impact of urban land use change on Digital Elevation Model and determine Flood Risk Susceptibility. Using analytical approaches like the land use change concepts coupled with mathematical modeling procedures promotes the explanatory power of satellite remote sensing-based measurements and methods in spatio-temporal environmental change assessments.

I think the analysis is logical and I found the results to contribute to the field of urban science, land use change impact and disaster risks assessment and reporting such as flood propagation in urban areas.

However, the manuscript could be improved by providing explicit information on the data used in terms of quality, plausibility and fitness for use. The manuscript does therefore need to be revised, I think it is a valuable contribution and it should be publishable following suitable revisions.

The authors modelled predicted urban growth but did not mention the accuracy of the land-cover footprint and information generated. Also, validation of the CA model, if not for 2022, which is the future but for the recent past time stamps (i.e. 2014 and 2018).

General questions

Digital Elevation Models (DEM) is a valuable dataset that provides input to natural hazard assessment and modeling such as flood risk analysis.

The quality of DEM sources and outputs is vital to the plausibility and fitness for use in different stages of natural hazard analysis and assessment of terrain and hydrological processes going on in any given area of interest. This is not mentioned in the manuscript and should be discussed. In fact, different DEMs, their resolution and terrain model derivatives differ due to data characteristics and the software used.

The critical assessment of DEM characteristics should be carried out and mentioned. In view, of these premises it is necessary to mention DEM sources, the standardization the DEM data underwent, software used. This will guide readers and potential up-takers of such ideas against false impression on DEM and its derivatives for use in decision making.

Line number reviews:

Lines 67-68: requires to be edited;

Line 24: replace susceptible with susceptibility

Author Response

Issue/line to improve

1.       Change the title to concise and shorter.

2.       Edit Line 24, 67-68

3.       Improve and Re-write the introduction  to the issue reference the effect of the change of DEM on the hydrologic characteristics, previous research (52-87)

4.       To improve Flowchart and similarity of methodology (133-195)

5.       Put the scale bar on every map, and adjust the details of the sharpness of the image.

6.       Describes the results from the CA model in basic translation, as well as adding the table data of every result from the model.(377-388) (410-437)

7.       What are the ways of studying with a past CA model and how we work differently? (420-471)

8.       Refer to how to get DEM data as well as processing methods. Software that use data-tuning DEM(475-494)

9.       The debate is based on the changes of DEM to the basin area. Our study confirms that DEM can be extracted to the extraction of independent variables that respond to a wide range of water-area changes.(508-535)

10.   Added the results table, Error matrix, Transition matrix, Transition probability matrix, Accuracy assessment, and the predicted LU in the year 2022.(707-732)

11.   Added Data source table (762-763)

12.   Add a reference list based on the content (788-803, 859-881)

Reviewer 2 Report

This paper is trying to predict the flood risk susceptibility based on DEM impacted by urbanization. There are three major concerns of this paper: 1) The assumptions and uncertainties in the model design need significant improvement. Each step is developed based on the previous invalidated models. Many ways can cause uncertainties and even bias in this paper, for example, the accuracy of mapping land-use, the accuracy of the CA model. All these uncertainties might lead your study to a completely different outcome. A session or sub-session of validation is highly recommended. 2) Data processing and data sources. Please list all the data sources used in this manuscript. 3) The scientific soundness of the results, and the whole session of “Discussion” is missing. The purpose of the discussion is to interpret and describe the significance of your findings in light of what was already known about the research problem being investigated and to explain any new understanding or insights that emerged as a result of your study of the prom. The discussion part can be considered as the most important part of your study. The title is long and confusing, for example, “Spatial modeling of build-up growth prediction and its impact”: Do you mean “spatial modeling’s” impact on DEM or “Built-up growth’s” impact on DEM? Please shorten the title, which can reflect your big picture. The methodology session has high similarity, the authors MUST fix this. All maps need the legend, scale bars (7 elements of a map)

Author Response

(The authors gave the same response as above.)

Round 2

Reviewer 1 Report

Accept in present form.

Might be good to include this article in the reference (https://www.mdpi.com/2072-4292/8/3/220). It fit into line 955 on urban growth modeling and the use of the CA model.